# Real-Time Feeding Behavior Monitoring by Electrical Penetration Graph Rapidly Reveals Host Plant Susceptibility to Crapemyrtle Bark Scale (Hemiptera: Eriococcidae)

**DOI:** 10.3390/insects13060495

**Published:** 2022-05-25

**Authors:** Bin Wu, Elizabeth Chun, Runshi Xie, Gary W. Knox, Mengmeng Gu, Hongmin Qin

**Affiliations:** 1Department of Horticultural Sciences, Texas A&M University, College Station, TX 77843, USA; bin.wu@tamu.edu (B.W.); fushe001@tamu.edu (R.X.); 2Department of Biology, Texas A&M University, College Station, TX 77843, USA; lizzie_chun1@tamu.edu; 3Department of Environmental Horticulture, University of Florida/IFAS North Florida Research and Education Center, Quincy, FL 32351, USA; gwknox@ufl.edu; 4Department of Horticultural Sciences, Texas A&M AgriLife Extension Service, College Station, TX 77843, USA; mgu@tamu.edu

**Keywords:** *Lagerstroemia*, crapemyrtle, *Acanthococcus lagerstroemiae*, crapemyrtle bark scale, CMBS, electrical penetration graph, EPG, rapid host confirmation, insect-plant interactions, EPGminer

## Abstract

**Simple Summary:**

Crapemyrtle bark scale (CMBS; *Acanthococcus lagerstroemiae*), an invasive polyphagous sap feeder in the United States, has spread across 16 U.S. states in less than two decades, posing potential risks to the Green Industry. Confirming the host range is crucial for effective pest management of invasive insects. However, host range confirmation relying on greenhouse or field trials is often inefficient and time-consuming. In this study, we used the electrical penetration graph (EPG) to monitor the stylet penetration of CMBS in plant tissue in real-time. An R programming-based application was developed to better characterize the insect EPG waveforms recorded by EPG. By analyzing EPG-based EPG parameters, we demonstrated that CMBS has difficulty accessing the phloem tissue (salivation and ingestion) of a resistant plant compared to a susceptible plant. Importantly, we hereby present CMBS typical feeding behaviors on susceptible and non-susceptible plants comparatively, which provides direct evidence for revealing unknown hosts rapidly.

**Abstract:**

Host range confirmation of invasive hemipterans relies on the evaluation of plant susceptibility though greenhouse or field trials, which are inefficient and time-consuming. When the green industry faces the fast-spreading threat of invasive pests such as crapemyrtle bark scale (*Acanthococcus lagerstroemiae*), it is imperative to timely identify potential host plants and evaluate plant resistance/susceptibility to pest infestation. In this study, we developed an alternative technology to complement the conventional host confirmation methods. We used electrical penetration graph (EPG) based technology to monitor the *A. lagerstroemiae* stylet-tip position when it was probing in different plant tissues in real-time. The frequency and relative amplitude of insect EPG waveforms were extracted by an R programming-based software written to generate eleven EPG parameters for comparative analysis between plant species. The results demonstrated that the occurrences of phloem phase and xylem phase offered conclusive evidence for host plant evaluation. Furthermore, parameters including the percentage of insects capable of accessing phloem tissue, time duration spent on initiating phloem phase and ingesting phloem sap, provided insight into why host plant susceptibility differs among similar plant species. In summary, this study developed a novel real-time diagnostic tool for quick *A. lagerstroemiae* host confirmation, which laid the essential foundation for effective pest management.

## 1. Introduction

As a new invasive sap-sucking insect initially in Texas [1,2], crapemyrtle bark scale [*Acanthococcus lagerstroemiae* (Hemiptera: Eriococcidae)] has spread across 16 U.S. states [3]. The potential ‘jumps’ of *A. lagerstroemiae* from its primary host [crapemyrtle (*Lagerstroemia* sp.)] to other economically important crops and native species pose a great risk to the Green Industry and ecosystems. Crapemyrtle is the best-selling deciduous flowering tree in the U.S.—3.03 million plants with a combined value of $69.57 million were sold in 2019 [4]. By virtue of its wide distribution and long blooming period, crapemyrtle provides excellent pollen sources for native and non-native bees in the U.S., especially when other resources are naturally scarce [5,6]. However, a heavy infestation of *A. lagerstroemiae* reduces crapemyrtle flowering/vigor and even causes branch dieback [7,8], which engenders a significant decrease in the sale number and value of crapemyrtle anticipated by the Green Industry [9].

Global warming, increasing human population densities, and escalating international trade of ornamental plants have been exacerbating invasive insects’ spreading [10,11]. The expanded distribution of *A. lagerstroemiae* could be serious due to its polyphagous feeding habit. Increased occurrence of *A. lagerstroemiae* has been reported on other plant species in the U.S., including native species [American beautyberry (*Callicarpa americana*), California loosestrife (*Lythrum californicum*), shrubby yellowcrest (*Heimia salicifolia*), St. Johnswort (*Hypericum kalmianum*)], economically important crops [Beer’s Black fig (*Ficus carica*), Chicago Hardy fig, ‘Fuji’ apple (*Malus domestica*), and pomegranate (*Punica granatum*)], and invasive plants [Japanese spiraea (*Spiraea japonica*) and Thunberg’s spiraea (*Spiraea thunbergii*)] [12,13,14,15,16,17,18]. These infestation reports have solidified the concerns that this fast-spreading invasive insect potentially threatened ecosystem stability and causes economic loss to the nursery and landscape industry [2]. Therefore, it is time-pressured and imperative to gain critical knowledge, especially the potential host plants, to control the spread of *A. lagerstroemiae*.

Time-saving and reliable approaches are in great need to rapidly reveal host plants for *A. lagerstroemiae*. The Green Industry is in need of management research of *A. lagerstroemiae* [9], where host evaluation research based on estimation of insect feeding performance is essential [19]. For chewing insects, estimating feeding performance and herbivory damage could be straightforward through short-period comparisons about the volume of consumed foliage [5,20]. However, conventional methods to estimate the feeding performance of sap-sucking insects typically rely on time-consuming tests regarding other indirect biological traits, such as hatching performance, growth rate, total fecundity, longevity, and the number of newly developed insects [12,13,21,22,23].

The stylet penetration of sap feeders in a plant could be a direct and crucial parameter for assessing host suitability through a real-time feeding monitor technique, electrical penetration graph (EPG) [24]. The EPG technique records variable voltage fluctuations regarding the style penetration activities over time termed EPG waveforms [25], tracking the position of the piercing-sucking insects’ stylet tip in different plant tissue in real-time. The EPG technology has been utilized as an important mainstay in elucidating insect-plant interactions (e.g., plant acceptance/resistance evaluation and pathogen transmission investigation) in the cases of aphids, leafhoppers, mealybugs, psyllids, and whiteflies [26,27,28,29,30,31,32]. To date, three main EPG waveforms have been identified and associated with the feeding behavior of sap-sucking insects based on histology correlation studies, namely waveforms C, E, and G. These three waveforms represent three main phases, respectively [28,29,33,34]. Waveform C represents the stylet pathway phase when gel saliva secretion, intercellular and intracellular stylet insertion or retraction, and short cell-puncturing occur [33]. Waveform E represents the phloem phase when phloem salivation and phloem ingestion occur [25]. Waveform G represents the xylem phase when xylem ingestion occurs [35]. Physical and chemical stimuli (including attractants, repellents, deterrents, nutritional traits, and defensive traits) at the plant surface or in epidermis/mesophyll, phloem tissue, and xylem tissue significantly influenced initial penetration time and total duration for each penetration activity spent at these three typical probing phases, respectively [36,37,38,39,40]. Such EPG parameters obtained by EPG recordings have proved critical for understanding host plant recognition, insect–plant interaction, and host plant resistance [41,42,43].

This study aims to establish an EPG-based system to monitor the stylet penetration of *A. lagerstroemiae* in the host plants. By comparing the EPG parameters of *A. lagerstroemiae* on a group of plants with variable host suitability, we have discovered key EPG parameters for host plant confirmation. These findings provide the Green Industry a fast diagnostic tool for quick *A. lagerstroemiae* host confirmation as well as guidelines on plant selection for pest management.

## 2. Materials and Methods

### 2.1. Insect Source and Plants

Colonies of *Acanthococcus lagerstroemiae* were established by attaching infested branches to healthy *Lagerstroemia limii* plants and maintained in a handmade chiffon mesh-covered cage (length, 58.0 cm; width, 58.0 cm; height, 50.0 cm) in 2019. The cage was placed in a CONVIRON^®^ (Controlled Environments Ltd., Winnipeg, MB, Canada) growth chamber [25 ± 1 °C, 60 ± 5% relative humidity (RH), and a photoperiod of 16 h light (L): 8 h dark (D)] at the Department of Biology, Texas A&M University. Crawling adult female of *A. lagerstroemiae* (length, 2.1 ± 0.7 mm; width, 1.2 ± 0.5 mm) obtained from the colony on detached twigs of the plants were used for EPG recordings.

A validated host *L. limii* [12] was used for characterizing the feeding behavior of *A. lagerstroemiae.* Plants used for comparing the feeding behavior by plant species were *L. limii* (positive control), *L. speciosa*, *L. indica* × *speciosa* ‘18096’, *Callicarpa acuminata*, *Ficus auriculata*, *F. pumila*, *F. tikoua*, and *Glycine max.* The crapemyrtle plants (*L. limii* and *L. speciosa*) were initially provided by North Florida Research and Education Center (Quincy, FL 32351, USA). The crapemyrtle hybrid ‘18096’ was selected from our crapemyrtle breeding program in the Department of Horticultural Sciences at Texas A&M University (College Station, TX 77843, USA). The three fig species and *C. acuminata* were initially provided by John Fairey Garden Conservation Foundation (Hempstead, TX 77445, USA). All these plants were propagated via cuttings and maintained in 1-quart plastic pots (The HC Companies, Twinsburg, OH 44087, USA) filled with potting substrate Jolly Gardener Pro-Line C/25 (Oldcastle Lawn and Garden Inc., Poland Spring, ME 04274, USA) in the greenhouse at 25 ± 5 °C, 50 ± 10% RH, and a photoperiod of 10.5: 13.5 (Light: Dark) hours (h).

### 2.2. Electrical Penetration Graph Recordings of Acanthococcus Lagerstroemiae Probing on Host Plant Lagerstroemia limii

To characterize the typical EPG waveforms, probing activities of individual *A. lagerstroemiae* adult female on *L. limii* were monitored in a Faraday cage using a Giga-8dd model direct current (DC) EPG amplifier with 1-gigaohm input resistance and an analog-digital (AD) conversion rate of 100 Hz (EPG Systems, Wageningen, The Netherlands). The EPG experiments were conducted in a climate-controlled room (25 ± 1 °C, 60 ± 5% RH, and a 16 h Light: 8 h Dark photoperiod) in the Department of Biology at Texas A&M University. After 24 h starvation, the dorsum of *A. lagerstroemiae* adult female was attached with a 2-cm long gold filament (diameter, 18.5 um) using water-based silver glue (EPG Systems, Wageningen, The Netherlands). The opposite end of the gold filament was glued with a copper wire (length, 3 cm; diameter, 0.2 mm), which was soldered with a brass nail being attached to an insect electrode. After inserting a substrate electrode into the potting substrate near the plant, the glued insect was placed on a stem of *L. limii*. The insects were monitored for at most 24 h per recording time, which was replicated 10 times using a new insect and a new plant per EPG recording.

According to how EPG waveforms were characterized in previous studies [25,28,32,33,44,45,46], the EPG recordings of *A. lagerstroemiae* were compared with the EPG waveforms of other sap-sucking insects. Based on the closely resembling waveform pattern, voltage level (extra- or intra-cellular), frequency (Hz), and relative amplitude (%), three typical EPG waveforms for *A. lagerstroemiae* on the host were successfully characterized.

### 2.3. EPG-Based Comparison of EPG Parameters among Different Plants

The probing activities of individual *A. lagerstroemiae* adult females on different plants were monitored as in the previous section to compare differences in EPG parameters. The test plants included *L. limii*, *L. speciosa*, *L. indica* × *speciosa* ‘18096’, *C. acuminata*, *F. auriculata*, *F. pumila*, *F. tikoua*, and *G. max*. The experiment was replicated 20 times for each plant species, and each replicate was conducted using a new insect and a new plant per EPG recording.

Based on the biological feeding activities, eleven EPG parameters of *A. lagerstroemiae* on each plant species were considered (Table 1). Three parameters (#1–#3) reflect the interactions between *A. lagerstroemiae* and plant surface, plant epidermis, and plant mesophyll, which include the percentage of individuals (among all tested insects) that started probing activities (#1), time from the start of the EPG recording until the successful initial probing (#2), total time of waveform C (sum of all penetration time spent by *A. lagerstroemiae* in stylet pathway, #3). Eight parameters reflect the interactions between *A. lagerstroemiae* and phloem or xylem tissue, which include the percentage of individuals showing waveform E1 (phloem salivation) in all tested insects (#4), time from the initial probing until the first E1 (#5), total duration of E1 (sum of all penetration time spent by *A. lagerstroemiae* in phloem salivation, #6), percentage of individuals having waveform E2 (phloem sap ingestion) in all tested insects (#7), time from the initial probing until the first E2 (#8), total duration of E2 (sum of all penetration time spent by *A. lagerstroemiae* in phloem ingestion, #9), percentage of individuals having waveform G (xylem ingestion) in all tested insects (#10), total duration of G (sum of all penetration time spent by *A. lagerstroemiae* in xylem ingestion, #11).

**Table 1 insects-13-00495-t001:** Electrical penetration graph parameters about feeding behavior of *Acanthococcus lagerstroemiae* on different plant species and statistical tests applied.

Electrical Penetration Graph Parameter	Statistical Test
Percentage of individuals starting probing (%)	GLM-Logistic
2. Time from EPG start until initial probing (min)	Welch’s ANOVA
3. Total duration of C ^z^ (min)	Welch’s ANOVA
4. Percentage of individuals showing E1 (%)	GLM-Logistic
5. Time from initial probing until first E1 (min)	Welch’s ANOVA
6. Total duration of E1 (min)	Welch’s ANOVA
7. Percentage of individuals showing E2 (%)	GLM-Logistic
8. Time from initial probing until first E2 (min)	Welch’s ANOVA
9. Total duration of E2	Welch’s ANOVA
10. Percentage of individuals showing G (%)	GLM-Logistic
11. Total duration of G	Welch’s ANOVA

^z^ C represents stylet pathway phase; E1 represents phloem salivation; E2 represents phloem ingestion; G represents xylem ingestion. See Figure 1.

**Figure 1 insects-13-00495-f001:**
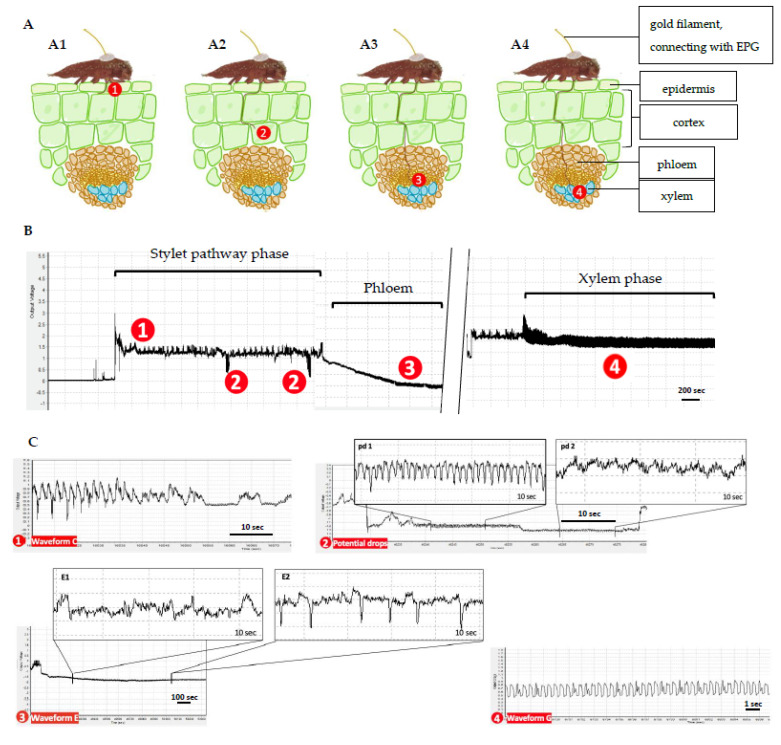
General scheme of typical EPG waveforms associated with specific probing activities of *Acanthococcus lagerstroemiae* on *Lagerstroemia limii*. (**A**) diagram shows different positions of *A. lagerstroemiae* stylet tips poking in plant tissue which were labelled as 1, 2, 3, and 4, respectively. (**B**) The general scheme shows specific EPG waveforms associated with the #1, #2, #3, and #4 positions of the stylet tip in the plant tissue, respectively. (**C**) ① Waveform C was detected when *A. lagerstroemiae* was probing intercellular part; ② Waveform potential drop (pd) was detected when the stylet tip punctured plant cells; ③ Waveform E1 was detected when intracellular stylet activity in mesophyll and phloem salivation occurred; Waveform E2, characterized by negative peaks, was detected when phloem sap ingestion occurred; ④ Waveform G was detected when xylem sap ingestion occurred.

### 2.4. Data Processing and Statistical Analysis

The Giga-8dd has the integrated AD converter on board with an USB output, and the data acquisition was visualized and recorded by Stylet+. Raw EPG recording per channel per experiment generally generates over 500 megabytes of data containing an enormous number of events/periods for each waveform. It requires a significant amount of time for a well-trained researcher to annotate the recordings second by second for accurately characterizing and interpreting the insect’s probing activities. Time-saving approaches to calculate values for each EPG waveform’s frequency and relative amplitude are much needed to yield a higher accuracy of identification and characterization of the feeding behavior of *A. lagerstroemiae*. Thus, we developed an R package named EPGminer with a user-friendly interface (epgdata.shinyapps.io/epgminer_app/, accessed on 23 March 2022) to streamline the characterization of typical EPG waveforms of *A. lagerstroemiae* by monitoring its probing activities (Appendix A). The EPG results were validated by comparing with our manual data annotations and the analysis results through the Excel Workbook [47,48]. In this study, we utilized EPGminer to semi-automatically identify and calculate frequency and relative amplitude for each waveform, which helps reduce human input (Table 2).

The R programming language version 4.1.0 on Windows 10 was used in conjunction with RStudio version 1.3.1073. The data were processed using several R packages. Both ggplot2 [49] and plotly [50] were used to generate visuals from R. Frequency and voltage values for each type of waveform within a dataset were calculated using functions wave_topfreq and wave_volts, respectively, from the EPGminer package. (1) The frequency value was calculated by using the Fast Fourier Transform (FFT) R implementation through the following methods. A center cut of the time-domain signal (length = 2^10^) was chosen as the most representative sample for each waveform. The sampling frequency (fs) was set to 100 Hz to match the data collection frequency. When calculating the single-sided amplitude, frequencies were selected up to the Nyquist frequency (fs/2 or 50 Hz in this case). All amplitudes were multiplied by two except for the DC component and the Nyquist frequency itself. According to the Nyquist criterion, to correctly reconstruct a repetitive sinusoidal wave with a unique amplitude, frequency, and phase, the sampling fs (100 Hz) should be greater than twice the highest frequency component in the FFT converted data to be sampled. Therefore, the highest frequency of the signal to be sampled was 50 Hz (fs/2). Additionally, the function for single sample FFT (epgminer::single_fft) applied a Hann (Hanning) window to reduce spectral leakage as well as a high pass filter of 0.25 Hz. Any frequencies below 0.25 Hz were regarded as noise and filtered out from the DC signal. In the converted data (frequency domain signal), the frequencies were aggregated into 1 Hz bins, with the largest amplitude-frequency chosen in each bin. The resulting 50 main frequencies were arranged in descending order of amplitude. The top 50 frequencies were chosen using the epgminer function mainfreqs, while the top one or single main constituent frequency for each waveform was then selected using the function wave_topfreq. (2) The voltage value for a certain waveform was calculated using the wave_volts function. The relative amplitude of volts (%RA) was calculated using the formula:$$\%\;\mathrm{RA}\;=\;\frac{\left|\left(\mathrm{amplitude}\;\mathrm{for}\;a\;\mathrm{certain}\;\mathrm{type}\;\mathrm{of}\;\mathrm{waveform}\;-\;\mathrm{amplitude}\;\mathrm{when}\;\mathrm{non}-\mathrm{probing}\right)\right|V}{5\;V}\;\times \;100\%$$

Data analysis for the comparison of EPG parameters by plant species, *L. limii* served as the positive control. Excel Workbook [47] and JMP^®^ 16 (SAS Institute Inc., Cary, NC, USA) were utilized to calculate and analyze all the parameters in Table 1. The parameters (#1, #4, #7, and #10) were analyzed by a generalized linear model fitted in a logistic function (GLM-Logistic) to test the effect of plant species on the percentage of *A. lagerstroemiae* that have a specific feeding activity among all tested insects. This method can be applied to small-sample clustered binary data [51,52]. For all the rest of the parameters in Table 1, Levene’s test and goodness-of-fit test (Shapiro–Wilk) were performed before analysis of variance (ANOVA) [31] to test homogeneity of variances and normality of data, respectively. The data related to the rest of the parameters were transformed using natural log transformation to improve model fitness appropriately [53], then analyzed by one-way ANOVA with Tukey’s Honestly Significant Difference (HSD) to test the effect of plant species on the time to initiate each penetration activity and the total duration of each EPG waveform. Otherwise, Welch’s ANOVA and Games–Howell post hoc (*α* = 0.05) tests were used to compare the difference in each mean value when the assumption of equal variances did not hold [54,55,56]. No individuals reached the phloem phase on *F. pumila*, *F. auriculata*, or *G. max*, these three species were excluded from the analyses regarding the parameters #5, #6, #8, and #9. *Ficus auriculata* was excluded from the #11 parameter comparison due to the same reason. The data were plotted using GraphPad Prism 9 (GraphPad Software, San Diego, CA, USA).

## 3. Results

### 3.1. Characterization of EPG Waveforms for A. lagerstroemiae Probing on a Validated Host, Crapemyrtle L. limii

Five typical EPG waveforms [C, potential drop (pd1and pd2), E1, E2, and G] have been characterized for *A. lagerstroemiae* using a validated host plant *L. limii* (Table 2 and Figure 1). Waveform F (often seen in EPG recordings of aphid or whitefly) was not clearly observed, indicating derailed stylet penetration is rare when *A. lagerstroemiae* was probing on *L. limii.* Comparing among hemipterans including mealybugs and aphids [25,28,32,33,44,45,46], the EPG waveforms of *A. lagerstroemiae* closely resemble that of mealybugs. Therefore, based on relative amplitude, voltage level (extra- or intracellular), and frequency, the EPG waveforms of *A. lagerstroemia* were characterized according to the standard waveform labeling method for mealybugs [28,32,44]. Each waveform is associated with a specific probing activity with detailed description listed below.

Waveform C, representing the stylet pathway phase (Figure 1C①), was detected when *A. lagerstroemiae* started probing and having extracellular stylet pathway activities (Figure 1(A1)). The frequency of waveform C was 1.80 ± 0.37 (Median ± SE) Hz, and the relative amplitude was 12.45 ± 1.13%.

Potential drops (Figure 1C②) were frequently observed during the stylet pathway phase (Figure 1(A2)). The voltage shifted to a lower level when the stylet tips punctured living plant cells. During the low intracellular voltage level, the potential drops were often clearly divided into potential drop 1 (pd1) and potential drop 2 (pd2) periods. The pd1 had a higher frequency (4.16 ± 0.48 Hz) than the pd2 (2.71 ± 0.16 Hz). The relative amplitude for the pd1 (20.12 ± 2.16%) and the pd2 (21.48 ± 2.38%) did not differ. In the end, the voltage returned to the extracellular level when the stylet bundles were withdrawn [28].

Waveform E complex (E1 and E2), representing the phloem phase (Figure 1C③), is often sequentially observed following the stylet pathway phase (Figure 1(A3)). The voltage level of the complex gradually dropped below zero, which was much lower than other waveforms. Waveform E1, potentially representing phloem salivation, had a similar frequency (1.52 ± 0.18 Hz) with waveform E2 (1.44 ± 0.24 Hz), potentially representing phloem ingestion. However, waveform E2 was composed of negative peaks (Figure 1C③). The relative amplitude for E1 and E2 was 21.20 ± 3.52% and 21.52 ± 4.55%, respectively.

Waveform G, representing the xylem phase (Figure 1C④), is correlated to xylem sap ingestion (Figure 1(A4)). Waveform G had a higher voltage level (extracellular) than waveform pd and E complex. Waveform G occurred at a frequency of 1.91 ± 0.20 Hz with a relative amplitude of 3.00 ± 0.81%.

### 3.2. Percentage of A. lagerstroemiae Initiated Stylet Pathway (Waveform C) and Time Spent by A. lagerstroemiae to Initiate and Stay in Stylet Pathway Differed on Different Plant Species

The percentage of the individuals that initiated probing significantly differed among the plant species (χ^2^ = 21.56; *df* = 7, 152.00; *p* = 0.0030; Table 3, #1). During the 24-h EPG recordings, 25–90% of the *A. lagerstroemiae* (*n* = 160) started probing on the respective test plants (Table 4, #1). 90% and 75% of the *A. lagerstroemiae* started probing on *L. limii* and *L. speciosa*, respectively, which was higher than 60% on *C. acuminata*, *F. tikoua*, and *G. max*, 55% on *F. pumila* and *L. indica × speciosa* ‘18096’. Only 25% of the individuals succeeded in starting probing on *F. auriculata* (Table 4, #1).

Plant species affected the time from onset of EPG recording until the initial probing (*F* = 2.71; *df* = 7, 33.56; *p* = 0.0243; Table 3, #2). After the initial probing, the total duration spent by *A. lagerstroemiae* in a stylet pathway phase (waveform C) differed significantly among the plant species (*F* = 10.43; *df* = 7, 36.46; *p* < 0.0001; Table 3, #3). Among the individuals that were detected with waveform C, *A. lagerstroemiae* spent a significantly longer time in the stylet pathway phase on *F. auriculata* and *G. max* than on *C. acuminata* (Table 4, #3).

### 3.3. Percentage of A. lagerstroemiae Accessed Phloem Tissue (Waveform E) and Time Spent by A. lagerstroemiae to Initiate Phloem Phase and Ingest Phloem Sap Differed on Different Plant Species

The phloem phase involves phloem salivation (E1) and phloem ingestion (E2) by *A. lagerstroemia.* The percentages of individuals having E1 (χ^2^ = 62.24; *df* = 7, 152; *p* < 0.0001; Table 3, #4) and the percentages having E2 (χ^2^ = 61.35; *df* = 7, 152; *p* < 0.0001; Table 3, #7) differed among plant species. There were 65% of individuals on the positive control plant *L. limii*, 55% on *C. acuminata*, 45% on *L. speciosa*, 40% on *F. tikoua*, and 20% the crapemyrtle hybrid ‘18096’ that reached the phloem with salivation. No individuals on the other three species (*F. pumila*, *F. auriculata*, and *G. max*) successfully targeted the phloem tissue (Table 4). The time from the initial probing until the first E1 (*F* = 4.94; *df* = 4, 14.47; *p* = 0.0102; Table 3, #5) and the time until the first E2 (*F* = 4.81; *df* = 4, 14.61; *p* = 0.0112; Table 3, #8) both differed among the plant species. Interestingly as shown in Table 4 (#5 and #8), *A. lagerstroemiae* spent an extended time to initiate phloem salivation and ingestion, respectively, on *L. speciosa* (632.29 min and 666.38 min, respectively) than on *C. acuminata* (476.30 min and 514.84 min) and *L. limii* (316.23 min and 341.81 min).

Among the individuals detected for waveform E, plant species did not affect their total duration of phloem salivation (*F* = 2.36; *df* = 4, 15.20; *p* = 0.0769; Table 3, #6). Plant species did significantly affect the total duration of phloem ingestion (*F* = 42.78; *df* = 4, 17.09; *p* < 0.0001; Table 3, #9). After reaching the phloem tissue, *A. lagerstroemiae* spent a significantly longer time in phloem sap ingestion (Table 4, #9) on the crapemyrtle hybrid ‘18096’ (307.66 min) and *L. limii* (200.92 min), which were at least three-fold longer than on *L. speciosa* (64.72 min), *F. tikoua* (26.49 min), and *C. acuminata* (24.60 min). No individuals had phloem salivation or phloem ingestion on *F. pumila*, *F. auriculata*, or *G. max*.

### 3.4. Percentage of A. lagerstroemiae Having Xylem Ingestion (Waveform G) Differed among Different Plant Species

The duration of xylem ingestion for *A. lagerstroemiae* ranges from 158.83 min to 408.49 min across different plant species (Table 4, #11). Although no significant difference was shown in the total penetration time spent by *A. lagerstroemiae* that entered xylem ingestion among the plant species (*F* = 2.25; *df* = 6, 24.60; *p* = 0.0716; Table 3, #11), the percentage of individuals having xylem ingestion differed (χ^2^ = 25.32; *df* = 7, 152.00; *p* = 0.0007; Table 3, #10). The highest percentage of the individuals having xylem ingestion was 60% on *C. acuminata*, followed by 50% on *L. limii*, 45% on both *L. speciosa* and the hybrid ‘18096’, 40% on *F. tikoua* and *F. pumila*. The lowest percentage of the individuals having xylem ingestion was 35% on *G. max*, while no individuals reached xylem tissue of *F. auriculata* in the 24-h EPG recordings (Table 4, #11).

## 4. Discussion

### 4.1. The First Assessment on the Feeding Behavior of A. lagerstroemiae Shows Clear Similarities and Differences in Characteristics Compared to Other Sap Feeders

From the perspective of stylet penetration activities, our study is the first report to elucidate the occurrences of phloem and xylem ingestion of *A. lagerstroemia*. *Acanthococcus lagerstroemiae* has very long and thin stylet bundles (Appendix A), which look similar to aphids [33], mealybugs [57], and whiteflies [58]. The general sequence and characteristics of the stylet penetration activities of *A. lagerstroemiae* should theoretically resemble these of other sap-sucking insects, which was validated through the EPG-based system in our study. These resembling characteristics involved the stylet insertion into the plant tissue with cell-puncturing during the stylet pathway phase, phloem salivation and ingestion, and xylem ingestion.

Our study demonstrated that *A. lagerstroemiae* has its different feeding characteristics from mobile sap-sucking insects. First, the mean time for *A. lagerstroemiae* to initiate the first probing and the first phloem ingestion was much longer, 238.90 min and 341.81 min on *L. limii* (Table 4, #2 and #8), as compared with 24.73 min and 80.67 min by aphids *Melanaphis sacchari* on *Sorghum bicolor* [52]. Second, the mean time spent by *A. lagerstroemiae* in phloem salivation on *L. limii* was 12.44 min (Table 4, #6), much longer than 1.38 min by aphids on *S. bicolor* [52]. Third, *A. lagerstroemiae* showed continuous phloem ingestion or xylem ingestion for several days without stylet retraction once it reached the suitable sieve element or xylem vessels. Only 1–2 occurrences of phloem ingestion or xylem ingestion were detected during a successful penetration by *A. lagerstroemiae*, compared to about six occurrences of phloem ingestion detected for aphids *Brevicoryne brassicae* [59].

As one of the strategies for *A. lagerstroemiae* to facilitate its penetration into the plant tissues by minimizing herbivory damage and avoiding plant defense, we hypothesized that the salivary secretions should contain pectinolytic enzymes to degrade constituents of plant cell walls and there might be a correlation between the enzyme activity and pectin content. Through agarose-pectin gel salivation substrate assays [60], we detected the pectinesterase from the saliva secreted by *A. lagerstroemiae* when the gels contained pectin, while no pectinesterase was detected when the gels were pectin-free (Appendix A). These results suggest that during stylet penetration, *Acanthococcus lagerstroemiae* might sense pectin in the plant tissues and react by pectinolytic enzymes secretion. However, neither the amount nor the size of the rings or halos differed between 0.1% pectin-added and 1.0% pectin-added substrates, indicating that the activity level of the pectinolytic enzymes secreted by *A. lagerstroemiae* was not associated with pectin content. It is challenging to speculate on the causation of this salivation for *A. lagerstroemiae*, as they are due to a combination of factors involved in the plant–insect interaction. For example, different processing capacities of chemosensors on the stylet tips might affect sap feeders’ abilities to initiate the first probing and phloem ingestion during plant selection and acceptance [61]. The chemosensors on the tip of antennae, labial tip chemoreceptors, and gustatory monitoring/discrimination are involved in the host selection and acceptance by aphids [32,62,63,64]. Thus, the characterization of feeding behavior via EPG system helps us better distinguish *A. lagerstroemiae* from aphids or other coccids in terms of insect–plant interaction. Further investigations on the fine structure of *A. lagerstroemiae* sensory organs related to its feeding behavior may explain why *A. lagerstroemiae* has a more sedentary way of feeding behavior.

### 4.2. Occurrences of Waveform E and G Detected by EPG Shows Vital Evidence to Unveil Host Plants of A. lagerstroemiae

Conventionally, the host of *A. lagerstroemiae* was determined by the presence of newly emerged infestation of the insect after attaching the infected branches to a test plant [65]. Due to the long life cycle of *A. lagerstroemiae*, our previous host confirmation and evaluation study took at least 25 weeks. We determined that *L. limii*, *L. speciosa*, *C. acuminata*, and *F. tikoua* as host plants for *A. lagerstroemiae*, versus *F. pumila* and *F. auriculata* as non-hosts [12,17]. In this study, we applied the real-time monitoring system to visualize *A. lagerstroemiae* probing activities on validated hosts versus non-hosts, which enables rapid host confirmation and evaluation for managing this pest in the future.

The comparative results of feeding parameters (Figure 2) showed that over 35% of the individuals in the tested *A. lagerstroemiae* could successfully ‘drink’ phloem sap and xylem sap from three validated host species (*L. limii*, *L. speciosa*, and *C. acuminata*) [12]. In contrast, no individual *A. lagerstroemiae* reached the phloem or xylem tissue on the non-hosts *F. pumila* and *F. auriculata* [17]. In most cases, with the assistance of endosymbionts, sap-sucking insects mainly acquire nutrients and energy from phloem and xylem for their growth and development [66,67,68]. It is reasonable to conclude that the plants where *A. lagerstroemiae* could not access phloem and xylem tissues should be non-hosts due to the failure to acquire sufficient nutrients.

Therefore, our EPG-based monitoring system provided convincing evidence to reveal hosts of *A. lagerstroemiae* in a timesaving manner. As a future research direction, testing other economically important crops and native species using this technology could better estimate this invasive insect’s potential threat to the Green Industry and ecosystems in the U.S.

### 4.3. Phloem Access and Ingestion Assist in Identifying Plant Resistance to A. lagerstroemiae

In our study, comparison results of the feeding parameters by plant species revealed close correlations of the phloem access and ingestion with the plant resistance levels (Table 4 and Figure 2). Among crapemyrtle species [12], *L. speciosa* was considered more *A. lagerstroemiae*-resistant than *L. limii* because the latter had higher number of ovisacs (peaking at 576 females in 17 weeks after the inoculation) compared to *L. speciosa* (peaking at 57 females in 19 weeks after the inoculation) in our greenhouse feeding trial. Our feeding parameter results showed that, compared with *L. limii*, a lower percentage of *A. lagerstroemiae* successfully reached the phloem phase on *L. speciosa*. After penetrating the epidermis, *A. lagerstroemiae* spent a significantly longer time to initiate the phloem salivation/ingestion with a huge decrease in the total duration of phloem ingestion on *L. speciosa* (Table 4).

Our results indicated that the poor *A. lagerstroemiae* performance (delays in insect development and the population reduction) potentially resulted from the delays in phloem-locating and difficulties in phloem ingestion. These phenomena could be due to higher plant resistance, involving morphological features and phytochemical adaptions (deterrence and repellents) in the mesophyll and phloem tissue [29,32,69,70]. No cultivated crapemyrtles have been reported to exhibit total resistance to *A. lagerstroemiae* [12]. Therefore, to improve *A. lagerstroemiae* resistance in crapemyrtles, it is essential to identify and integrate the main plant resistance mechanism by investigating the correlations between the *A. lagerstroemiae* feeding behavior and nutritive and defensive traits in phloem sap from resistant and susceptible crapemyrtles.

In this study, difficulties in reaching phloem tissue and a decreased duration in phloem ingestion were indicated by feeding parameters when *A. lagerstroemiae* was feeding on the resistant plant *L. speciosa*, which improved our understanding of the interaction between plant and *A. lagerstroemiae*. To further decipher the plant resistance mechanisms against *A. lagerstroemiae*, high-resolution micro-computed tomography would be needed to demonstrate similarities and differences in how *A. lagerstroemiae* penetrates a resistant plant and a susceptible plant. With the integration of ‘omic’ research approaches (e.g., transcriptomics, metabolomics, and QTLomics), comparative analysis on phloem tissue between *L. limii* and *L. speciosa* will provide a holistic understanding of plant resistance to *A. lagerstroemiae.* Such endeavors will increase the breeding efficiency for desired plant traits, benefiting academia and industry communities.

## 5. Conclusions

In summary, this is the first report to characterize the typical EPG waveforms of *A. lagerstroemiae* by the EPG-based monitoring system. Based on the analyses of eleven feeding parameters, the occurrences of waveform E and G could be convincing evidence to reveal host plants of *A. lagerstroemiae* rapidly. After initiating the first probing, *A. lagerstroemiae* spent more time locating the phloem tissue and a shorter time for phloem sap salivation and ingestion on resistant plant *L. speciosa* than on susceptible plant *L. limii*. This work allows us to clearly distinguish the similarities and differences in feeding features between *A. lagerstroemiae* and other sap-sucking insects according to the typical EPG waveforms. With the help of EPG technologies, this work improves our understanding of the correlations between plant resistance/susceptibility and feeding parameters of *A. lagerstroemiae*.

## Figures and Tables

**Figure 2 insects-13-00495-f002:**
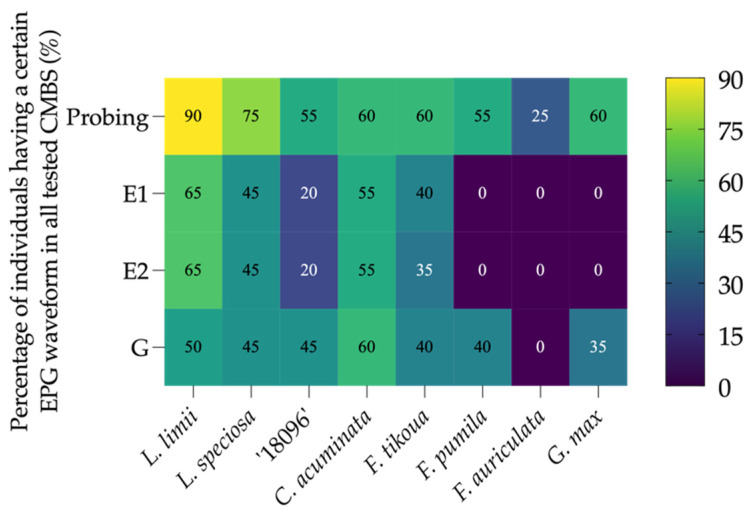
Percentage of individuals having a certain EPG waveform in all tested *Acanthococcus lagerstroemiae* samples among eight plant species.

**Table 2 insects-13-00495-t002:** Characteristics of the EPG waveforms recorded during *Acanthococcus lagerstroemiae* probing on *Lagerstroemia limii*.

EPG Waveform		Waveform Characteristics				Correlations
		Voltage Level	Frequency (Hz)		Relative Amplitude (%) ^z^	Activities Assigned for Similar Waveforms in Other Hemipterans
			Min–Max	Median ± SE	Median ± SE	
C ^y^		Extracellular	0.98–4.57	1.80 ± 0.37	12.45 ± 1.13	Sheath salivation and other intercellular stylet pathway
pd	pd1	Intracellular	2.71–6.62	4.16 ± 0.48	20.12 ± 2.16	Short cell punctures
	pd2	Intracellular	1.50–3.14	2.71 ± 0.16	21.48 ± 2.38	
E1		Intracellular	0.49–2.05	1.52 ± 0.18	21.20 ± 3.52	Phloem salivation
E2		Intracellular	0.68–2.78	1.44 ± 0.24	21.52 ± 4.55	Phloem sap ingestion
G		Extracellular	1.46–3.12	1.91 ± 0.20	3.00 ± 0.81	Xylem sap ingestion

^z^ Relative amplitude (%) = |(mean of amplitude for each waveform-mean of amplitude for non-probing)|V/5 V × 100%. C ^y^ represents stylet pathway phase; E1 represents phloem salivation; E2 represents phloem ingestion; G represents xylem ingestion.

**Table 3 insects-13-00495-t003:** GLM-Logistic and Welch’s ANOVA results about EPG parameters of *Acanthococcus lagerstroemiae* on eight different plant species.

Electrical Penetration Graph Parameter	$F\;\mathrm{or}\;\chi^{2}$	Num *df*, Denom *df*	*p*-Value
Percentage of individuals starting probing (%)	21.56	7, 152.00	0.0030
2. Time from EPG start until initial probing (min)	2.71	7, 33.56	0.0243
3. Total duration of C ^z^ (min)	10.43	7, 36.46	<0.0001
4. Percentage of individuals showing E1 (%)	62.24	7, 152	<0.0001
5. Time from initial probing until first E1 (min)	4.94	4, 14.47	0.0102
6. Total duration of E1 (min)	2.36	4, 15.20	0.0992
7. Percentage of individuals showing E2 (%)	61.35	7, 152	<0.0001
8. Time from initial probing until first E2 (min)	4.81	4, 14.61	0.0112
9. Total duration of E2	42.78	4, 17.09	<0.0001
10. Percentage of individuals showing G (%)	25.32	7, 152	0.0007
11. Total duration of G	2.25	6, 24.60	0.0716
12. Percentage of individuals starting probing (%)	21.56	7, 152.00	0.0030
13. Time from EPG start until initial probing (min)	2.71	7, 33.56	0.0243
14. Total duration of C ^z^ (min)	10.43	7, 36.46	<0.0001
15. Percentage of individuals showing E1 (%)	62.24	7, 152	<0.0001
16. Time from initial probing until first E1 (min)	4.94	4, 14.47	0.0102
17. Total duration of E1 (min)	2.36	4, 15.20	0.0992
18. Percentage of individuals showing E2 (%)	61.35	7, 152	<0.0001
19. Time from initial probing until first E2 (min)	4.81	4, 14.61	0.0112
20. Total duration of E2	42.78	4, 17.09	<0.0001
21. Percentage of individuals showing G (%)	25.32	7, 152	0.0007
22. Total duration of G	2.25	6, 24.60	0.0716

^z^ C represents stylet pathway phase; E1 represents phloem salivation; E2 represents phloem ingestion; G represents xylem ingestion.

**Table 4 insects-13-00495-t004:** Electrical penetration graph parameters of *Acanthococcus lagerstroemiae* probing on different plant species.

Electrical Penetration Graph Parameter ^z^	Plant Type							
	*Lagerstroemia limii*	*Lagerstroemia speciosa*	*Lagerstroemia indica* × *speciosa* ‘18096’	*Callicarpa acuminata*	*Ficus tikoua*	*Ficus pumila*	*Ficus auriculata*	*Glycine max*
Percentage of individuals starting probing (%)	90.00	75.00	55.00	60.00	60.00	55.00	25.00	60.00
2. Time from EPG start until initial probing (min)	238.90 (181.85–313.77) ^y^ ab ^x^	486.85 (361.06–656.32) a	352.85 (248.90–500.52) ab	324.00 (231.96–500.52) ab	257.28 (184.18–359.48) b	404.52 (285.33–573.44) ab	397.29 (236.71–666.91) ab	453.25 (324.49–633.12) ab
3. Total duration of C (min)	483.26 (361.28–646.55) abc	531.79 (386.70–731.91) bc	475.07 (327.23–689.29) abc	502.23 (352.01–716.70) c	749.61 (525.14–1070.26) abc	664.18 (457.90–962.62) bc	1033.65 (595.65–1793.02) a	795.49 (557.61–1135.30) ab
4. Percentage of individuals having E1 (%)	65.00	45.00	20.00	55.00	40.00	0.00	0.00	0.00
5. Time from initial probing until first E1 (min)	316.23 (253.32–394.51) b	632.29 (484.28–825.22) a	454.53 (304.80–677.67) ab	476.30 (374.15–605.86) b	473.69 (356.97–628.07) ab	- ^w^	-	-
6. Total duration of E1 (min)	12.44 (7.32–21.14) a	30.67 (16.21–58.03) a	46.80 (17.99–121.80) a	34.32 (19.28–61.10) a	21.56 (10.96–42.39) a	-	-	-
7. Percentage of individuals having E2 (%)	65.00	45.00	20.00	55.00	35.00	0.00	0.00	0.00
8. Time from initial probing until first E2 (min)	341.81 (279.40–418.07) b	666.38 (523.04–849.51) a	506.57 (352.36–728.99) ab	514.84 (413.50–640.76) b	540.86 (411.03–711.70) ab	-	-	-
9. Total duration of E2	200.92 (120.18–335.85) a	64.72 (34.90–120.02) b	307.66 (226.91–416.82) a	24.60 (14.07–43.20) c	26.49 (13.15–53.33) bc	-	-	-
10. Percentage of individuals having G (%)	50.00	45.00	45.00	60.00	40.00	40.00	0.00	35.00
11. Total duration of G (min)	158.83 (107.61–234.55) a	387.55 (232.65–584.44) a	320.62 (212.72–483.31) a	375.50 (263.18–535.74) a	330.35 (227.91–478.98) a	408.49 (264.32–631.22) a	-	199.07 (75.87–522.52) a

^z^ C represents stylet pathway phase; E1 represents phloem salivation; E2 represents phloem ingestion; G represents xylem ingestion. ^y^ Natural log transformation as log (duration of each EPG parameter) was conducted before data analysis. The inverse-transformed means with the inverse-transformed 95% confidence interval (CI) for each EPG parameter were presented. ^x^ Means with 95% CI followed by different lowercase letters within a row are different by the Games–Howell test (*α* = 0.05). ^w^
*Ficus pumila*, *Ficus auriculata*, and *Glycine max* were excluded from the analyses regarding the #5, #6, #8, and #9 parameters. In addition, *F. auriculata* was excluded from the analysis regarding the #11 parameter.

## Data Availability

The data presented in this study are available within the article and Appendix A.

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
