# Peer review of "Real-Time Feeding Behavior Monitoring by Electrical Penetration Graph Rapidly Reveals Host Plant Susceptibility to Crapemyrtle Bark Scale (Hemiptera: Eriococcidae)"

_insects, 2022, doi:10.3390/insects13060495_

Round 1

Reviewer 1 Report

This is an excellent paper and I congratulate the authors for this good study!!

This is an important study which develops an alternative technology to complement the conventional host plant resistance methods in entomology and particularly well adapted to sap feeders. The well known technology used, i.e. electrical penetration graph (EPG), is a good one to monitor electrically the stylet-tip position of the insects in real-time
and to demonstrate clearly that it is a very useful tool to follow the stylets pathways of such insect into the plant tissues when it is feeding on plants. The use of EPG allowed them to develop a novel real-time diagnostic tool for quick host confirmation preference and resistance of the pest insect, which laid the essential foundation for effective pest management, particularly useful to understand the host plant resistance process.

Although this technology is commonly used on such sap feeders insects, the authors used it in an original manner in order to provide convincing evidence to characterize the more of less preferred hosts of A. lagerstroemiae in a timesaving manner. Also, this is the first report to characterize the typical feeding waveforms of A. lagerstroemiae in the literature by the EPG-based monitoring system. 

With the help of their EPG technology used in their study, the authors confirm (because this was highlighted before their study in the literature) that EPG can help significantly to understand the correlations between plant resistance/susceptibility and feeding parameters of such sap feeder insects. Also this study gives a future research direction, testing other economically important crops and other species using this technology. 

The introduction and discussion were well argued by appropriate references

The figures illustrates well the EPG technology and what kind of information the electrical signals generated by EPG gives on the stylet-tip position of the insects into plant tissues. I particularly liked the Figure 1 for that. The tables, although not obviously easy to read, are necessary to get the entire feedback of the different EPG waveforms obtained on different plants, and then well summarized somehow at Figure 2. 

Author Response

We greatly appreciate the positive comments from reviewer 1. 

Reviewer 2 Report

The article suggests to provide very interesting new EPG waveform identification and data processing techniques. Though methodologically breaking news (!), the article needs some revision to clarify backgrounds, used methods, and left open questions.

Author Response

Reviewer 2’s comments are provided in two files. We have made all changes suggested by the reviewer on the manuscript. These changes are marked up with tracking changes in the revised manuscript. Here we provide point-by-point responses to the reviewer’s comments.

Article

The article suggests to provide very interesting new EPG waveform identification and data processing techniques. Though methodologically breaking news (!), the article needs some revision to clarify backgrounds, used methods, and left open questions.

Response: We thank the reviewer’s encouraging comments. We have revised the manuscript according to the reviewer’s comments.

Terminology

Feeding  In general feeding in many insects is generally used. In combination with ‘behavior’ it seems OK for insects with piercing sucking mouthparts as well. However as a separate activity it may be confusing when ‘probing’ activities in general, or more specific ‘ingestion’ is meant. In ‘feeding waveforms’, ‘EPG waveforms’ will be better. I changed and put comments on the use of ‘feeding’ up to line 150. Please check ‘feeding’ in the remainder of the manuscript.

Response: The authors agreed with the reviewer’s suggestions and replaced ‘feeding waveforms’, ‘feeding parameters’, and ‘feeding activities’ with ‘EPG waveforms’ and ‘EPG parameters’, and ‘probing activities’, respectively,  in the entire manuscript (Lines 21, 22, 33, 34, 35, 36, 100, 101, 105, 107, 134, 136, 151, 155, 157, 158, 160, 190, 191, 196, 228, 240-241, 249, 258, 260, 285-286, 289, 291, 351, 355, 357, 358, 423, 476, 483-484).

Material and methods

Line 117    Nothing is mentioned about the sessile life of adult females and any difficulties that might be expected by removing and new settling of them. Normally these stages will not move and new stylet insertion (for EPG recording here) does not occur.

L140: As far as I can see and is shown in https://extensionentomology.tamu.edu/ ”only the crawler (L1) seems mobile and is the only stage that can disperse” (mainly by wind). So, how disturbing is moving adult females, how easy they resume probing? Refer to published articles!

Response: Indeed, once settled, adult females rarely move. However, the authors found that insects including crawlers (L1 and L2), and adult females all would retract their stylets and become mobile when the segment of infested stem was cut away from the whole plant. This observation allowed the authors to collect intact and suitable adult females for EPG experiment.

Since there are no prior publications documenting this observation, we added a sentence in L117-120 to clarify how we obtained adult females: “Crawling adult females of A. lagerstroemiae (length, 2.1 ± 0.7 mm; width, 1.2 ± 0.5 mm) obtained from the colony on detached twigs of the plants were used for EPG recordings.”

Line 182    May be some information would be helpful here to provide the raw data file format: e.g.

Stylet+ binary 1 hour, 1 channel data files; Windaqlite has multichannel multi-hour format, which is suggested not suitable?

Response: We clarified the data format by including the following information in the revision: The Giga-8dd has the integrated AD converter on board with an USB output, and the data acquisition was visualized and recorded by Stylet+(L185-186).

Specific comments:

Comment 1.   Electrical penetration graph parameters (Table 1)

Are the 11 EPG parameters (variables) a maximum number from the (most recent) version of the epg-189 data.shinyapps.io/epgminer_app/ application? And - as a consequence the - only variables were compared with the (Spanish) Excel Workbook (check the latest version of it!) ?

Has the epgminer app used the same calculation strategy for each of the 11 parameters as this Workbook (as is suggested on line 193).

If so, to prevent confusion this should be mentioned clearly in text!

Nevertheless, for

#1 Line 146 says ‘recording… was replicated 10 times’, whereas on line 164 says the ‘percentage among all tested insects so that the 90% (Table 4) of the 20 replicates (line 158) will be 18?

#2 Does this concern the average (or median?) time to 1st C (t>1C) in:

  • only the 18 replicates with C?
  • all 20 replicates, and

> if no C then these are discarded

> if no C then the complete recording time is taken as t>1C ?

Etc.

Also in Discussion, Fig2: “EPG waveform(s) in all tested … samples”, is that in all 20 or in the 18with C?

Statistics:  The Table 1 statistics are not my expertise. So, I will leave commenting these to others.

Response: Thanks for pointing this out. The manuscript mentioned in L196-200 that EPGminer was mainly used to assist with characterization of EPG waveforms- identify and calculate frequency and RA for each waveform. It was not used for comparing the 11 parameters among different plant species in this manuscript. The 11 parameters were calculated and comparatively analyzed using the Excel Workbook and JMP, as mentioned in L228-230.

Comment 2  Relative amplitude

It seems that your formula:

% Ra =                                                                                                                     x 100%

uses for each waveform its absolute peak to peak value |max - min| subtracted by the absolute np noise amplitude, i.e. |waveform amplitude| - |noise amplitude| ; correct?

However, waveform amplitudes are voltage supply (Vs) dependent, at least the R components in the waveform signal. Division by 5V in your formula doesn't make much sense, therefore.

The shifts in voltage level (extra-/intra-cellular) of pd waveforms (aphid), G waveform (mainly), and E1 waveforms are emf originated and independent of the Vs adjustment values whereas the amplitude of the waveform E2 downward spikes will increase when Vs is adjusted more negative (Fig. 1C-3) and decrease when Vs is adjusted more positive. Thus in each recording (or even within a recording) the %Ra can be different.

Response: In the formula, the numerator we used is ‘|waveform amplitude - noise amplitude|’. It is not ‘|waveform amplitude| - |noise amplitude| ’as interpreted by the interviewer. The authors subtracted the np noise amplitude from the measured amplitude for each waveform, so all recorded amplitudes were comparable. The absolute value was then used as the numerator in the formula for %RA calculation. Since the noise amplitude can be positive or negative, ‘|waveform amplitude| - |noise amplitude|’ could introduce unintended errors. We therefore did not use ‘|waveform amplitude| - |noise amplitude|’ in the formula.

The authors agreed that waveform amplitudes are mainly Vs dependent. In the EPG equipment we used for this study, Style+ scale range limits were  5Volt (recording display). The power supply was 5V. The power supply output voltage was a constant. The amplitudes for waveform E2 downward spikes in all recordings for the same plant species were relatively consistent. In terms of the %RA calculation, the highest voltage of waveform A or 5V was considered to be the 100% relative amplitude in studies using similar settings (see other insect publications * below). In this study, the authors selected 5V as the highest voltage for the RA calculation for each waveform and used it as the denominator in the formula.

* Cornara, D., Garzo, E., Morente, M., Moreno, A., Alba-Tercedor, J. and Fereres, A., 2018. EPG combined with micro-CT and video recording reveals new insights on the feeding behavior of Philaenus spumarius. PLoS One, 13(7), p.e0199154. https://doi.org/10.1371/journal.pone.0199154

Dancewicz, K., GabryÅ›, B., Morkunas, I. and Samardakiewicz, S., 2021. Probing behavior of Adelges laricis Vallot (Hemiptera: Adelgidae) on Larix decidua Mill: description and analysis of EPG waveforms. Plos one, 16(5), p.e0251663. https://doi.org/10.1371/journal.pone.0251663

Comment 3  Other waveform characteristics

Waveforms pd1 and pd2 are different from aphids and mealybugs (3 sub-phases) and need more characterizing, and comparison with pds by these other insects. Here the later discussed occurrence (number/time C; min).

Waveform F (aphids, whiteflies, mealybugs) seem not observed in your scales? If so, this could be mentioned.

Waveform frequencies

Suggested that < 0.25Hz is neglected:

  • except for the pd voltage level shifts (0Hz)?
  • waveform B - clearly shown in your Fig. 1C-1 - has 0.2Hz frequency in aphids. May be B has a higher frequency in scales (?) Although waveform B is considered as such but are lumped with A and C and labeled as C in waveform analysis of aphid signals.
  • Time scales in Fig. 1B and 1C are out of focus: a time bar in these figures will be helpful.

Response: These are insightful suggestions. The crapemyrtle aphid (Sarucallis kahawaluokalani) is another common and conspicuous insect pest of crapemyrtle in the U.S*. Mealybugs were also seen frequently visiting crapemyrtles.  Our future work will include investigations to compare the EPG waveforms/parameters among CMBS, crapemyrtle aphids, and mealybugs would be beneficial for developing integrated pest management of sap-sucking insects.

* Herbert, J.J., Mizell III, R.F. and McAuslane, H.J., 2009. Host preference of the crapemyrtle aphid (Hemiptera: Aphididae) and host suitability of crapemyrtle cultivars. Environmental entomology38(4), pp.1155-1160. https://doi.org/10.1603/022.038.0423

We added a sentence to mention no waveform F was observed (L253-255). “Waveform F (often seen in EPG recordings of aphid or whitefly) was not clearly observed, indicating derailed stylet penetration is rarewhen A. lagerstroemiae was probing on L. limii.”

When waveform frequencies were calculated, signals < 0.25Hz were eliminated in our analysis (L217-219). As noted by the reviewer, the software Stylet+ used in this study lumped waveforms A, B and C together as waveform C as well.

As the reviewer suggested, the scale bars were added in Figure 1-B and 1C.

Comment 4 Discussion

Line 367-371        The longer initial period (time to 1st probe) seems mainly due to the bark and branch cuticle features by the scales compared to the leaves and stems cuticle structures of herbaceous host plants in most EPG studies on aphids, whitefly, and mealybugs. Also, may the 1h starvation was not enough for the adult females to rearrange their stylet properly after their interrupted sessile life.

Response: The authors thank the reviewer’s insightful suggestions. Future work will be conducted to address the differences in feeding characteristics among CMBS and other sap-sucking insects.

In this study, 24 hrs, not 1hr, starvation was used (L142).

Line 371-373        There is no obvious cause to be mentioned for the much longer E1 periods in Melanaphus (and other?) aphids., indeed.

Response: In the cited reference, there is no obvious or direct causation mentioned for the much longer E1 periods in Melanaphis sacchari. However, it was found that the duration of E1 was longer and the number of E2 was reduced on a resistant melon line AR 5 z. And high levels of callose deposited around most of the stylet sheaths when the melon aphids were feeding on the resistant line AR 5 y. Therefore, phloem sieve element occlusion by callose and P-proteins might cause the longer E1 period x.

In our manuscript, the reason that the authors compared EPG parameters between CMBS and aphids is to clarify the different feeding characteristics of CMBS from another sap-sucking insect.  

Z Klingler, J., Powell, G., Thompson, G.A. and Isaacs, R., 1998. Phloem specific aphid resistance in Cucumis melo line AR 5: effects on feeding behaviour and performance of Aphis gossypii. Entomologia Experimentalis et Applicata, 86(1), pp.79-88. https://doi.org/10.1046/j.1570-7458.1998.00267.x

Y Shinoda, T., 1993. Callose reaction induced in melon leaves by feeding of melon aphid, Aphis gossypii Glover, as possible aphid-resistant factor. Japanese Journal of Applied Entomology and Zoology. 37, pp. 145-152.

X Walker, G.P., 2021. Sieve element occlusion: Interactions with phloem sap-feeding insects. A review. Journal of Plant Physiology, p.153582. https://doi.org/10.1016/j.jplph.2021.153582

Line 373-377        It seems likely that the sessile female scale occurrence enables continuous phloem ingestion periods opposite to the mobile features of aphids and whitely adults. Whitefly nymphs are more comparable (Lei et al. 1995)

Response: The authors agree with the reviewer. The word “other” in the sentence in L374-375 has been changed to “mobile” to highlight the difference between the sessile CMBS and crawling aphids.

Line 378      “the insect’s stylet bundles were coated with secretions of salivary sheath’ rather unlikely this saliva coating is provided by the insect before probing. Sure, during probing inside the plant tissue, but how could it remain on the stylets when the stylets were dragged out of the tissue? The salivary sheath normally remains inside the plant when the insect withdraws its stylets, either naturally or by removing using a brush or pulling the insect attached gold wire. Anyhow, it sounds strange, and the picture (poorly focused) can hardly be considered as a convincing proof.

Response: We agree with the reviewer. To avoid confusion, Figure S2 has been removed from the revised manuscript and renumbered ‘Figure S3’ as ‘Figure S2’.

The role of pectin as a structural obstacle to stylet penetration seems not very important as such. It is rather weak substance in comparison to lignin and (hemi)cellulose. May be it plays a role in the cellulose-hemicellulose fiber coherence. See also Cherqui & Tjallingii (1999), Urbanska et al. (1998) and many more recent publications on salivary enzymes.

Response:  We agree with the reviewer. In this study, the authors validated the presence of pectinesterase in the salivary secretions by CMBS. Further investigation would be needed to investigate whether any other salivary enzymes are present in the saliva involved in the CMBS-plant interactions.

Salivary secretions contain various enzymes. Polyphenol oxidase (PPO) and peroxidase (Px) in the salivary secretions, as Urbanska et al. (1998)y mentioned, play roles in plant phenolics metabolization (food digestion). Besides that, cell-degrading enzymes enable insect stylet to probe into the plant tissue with little damage. For example, amylase, pectinase, cellulase, and pectinesterase. Allelochemical-detoxifying enzymes detoxify plant chemicals and enable insects to circumvent the plant defense against herbivory attack, such as glutathione peroxidase, laccase, and trehalasex.

Physical and chemical features of plant, including pectin, impact the insect-plant interactions. Calatayud et al. found that pectinolytic enzymes in the salivary secretions by mealybug involved in the degradation of the cassava cell wallz.

z Calatayud, P.A., Boher, B., Nicole, M. and Geiger, J.P., 1996. Interactions between cassava mealybug and cassava: cytochemical aspects of plant cell wall modifications. In Proceedings of the 9th International Symposium on Insect-Plant Relationships (pp. 242-245). Springer, Dordrecht.

y Urbanska, A., Tjallingii, W.F., Dixon, A.F. and Leszczynski, B., 1998. Phenol oxidising enzymes in the grain aphid's saliva. Entomologia Experimentalis et Applicata, 86(2), pp.197-203. https://doi.org/10.1046/j.1570-7458.1998.00281.x

x Sharma, A., Khan, A.N., Subrahmanyam, S., Raman, A., Taylor, G.S. and Fletcher, M.J., 2014. Salivary proteins of plant-feeding hemipteroids–implication in phytophagy. Bulletin of entomological research, 104(2), pp.117-136. https://doi.org/10.1017/s0007485313000618

Line 395-404    Wensler demonstrated the lack of chemo receptors; only mechanoreceptors on the stylet tips! Internal cibarial chemoreceptor may change salivary contents after sap sampling, antennal receptors will certainly not. It seems unlikely that the differing pd durations and frequencies can be explained by the sensory inputs, a different evolutionary history will be more likely.

Response: The authors agree with the reviewer that insect evolutionary differences could contribute to the diversity in the EPG recordings among sap-sucking insects. And the authors deleted this sentence (L405-407) to make the whole context focus on discussing the insect-plant interaction.

Indeed, the tip of the labium of the aphid only possesses mechanoreceptor z,y. However, the labium of the whitefly has contact chemoreceptors x, and the labial tip of the mealybug has contact and olfactory chemoreceptorsw. The mealybug (Hemiptera: Pseudococcidae) and CMBS (Hemiptera: Eriococcidae) are in the Coccomorpha group. It is reasonable to assume that the labium of CMBS possesses mechano- and chemoreceptors. The physical and chemical characteristics of the plant are likely involved in CMBS-plant interaction (L403-405). These are very interesting topics for future research.

z Wensler, R.J., 1977. The fine structure of distal receptors on the labium of the aphid, Brevicoryne brassicae L.(Homoptera). Cell and Tissue Research, 181(3), pp.409-422. https://doi.org/10.1007/BF00223115

y Tjallingii, W.F., 1978. Mechanoreceptors of the aphid labium. Entomologia experimentalis et applicata, 24(3), pp.731-737. https://doi.org/10.1111/j.1570-7458.1978.tb02837.x

x Walker, G.P. and Gordh, G., 1989. The occurrence of apical labial sensilla in the Aleyrodidae and evidence for a contact chemosensory function. Entomologia experimentalis et applicata, 51(3), pp.215-224. https://doi.org/10.1111/j.1570-7458.1989.tb01232.x

w Le Rü, B., Renard, S., Allo, M.R., Le Lannic, J. and Rolland, J.P., 1995. Ultrastructure of sensory receptors on the labium of the cassava mealybug, Phenacoccus manihoti Matile Ferrero. Entomologia experimentalis et applicata, 77(1), pp.31-36. https://doi.org/10.1111/j.1570-7458.1995.tb01982.x

4.3. Phloem access

Line 434 “females in 17 weeks after the inoculation”. Are the inoculations due to migrating L1 crawlers only, or have moving later stages and adult females caused these populations as well?

Response: The sentence in the revision L433-444 reads, “…had higher number of ovisacs (peaking at 576 females in 17 weeks after inoculation) …” In the cited references, the “females” referred to the number of newly developed ovisacs by gravid females on a plant after the inoculation. Crawlers or later stages insects were not counted for the female number.

Here are the citations. 

https://doi.org/10.1653/024.102.0129

https://doi.org/10.3390/insects11070399

https://doi.org/10.3390/insects12010006

https://doi.org/10.21273/HORTTECH04897-21
